# A randomised controlled trial of a duodenal-jejunal bypass sleeve device (EndoBarrier) compared with standard medical therapy for the management of obese subjects with type 2 diabetes mellitus

Michael Alan Glaysher,[1] Aruchuna Mohanaruban,[2] Christina Gabriele Prechtl,[3] Anthony P Goldstone,[4] Alexander Dimitri Miras,[5] Joanne Lord,[6] Navpreet Chhina,[7] Emanuela Falaschetti,[8] Nicholas Andrew Johnson,[9] Werd Al-Najim,[10,11,12] Claire Smith,[8] Jia V Li,[13] Mayank Patel,[14] Ahmed R Ahmed,[2] Michael Moore,[15] Neil Poulter,[8] Stephen Bloom,[16] Ara Darzi,[17] Carel Le Roux,[18] James P Byrne,[19] Julian P Teare[20]

For numbered affiliations see end of article.

**Correspondence to**
Michael Alan Glaysher;
michaelglaysher@me.com

## ABSTRACT

**Introduction** The prevalence of obesity and obesity-related diseases, including type 2 diabetes mellitus (T2DM), is increasing. Exclusion of the foregut, as occurs in Roux-en-Y gastric bypass, has a key role in the metabolic improvements that occur following bariatric surgery, which are independent of weight loss. Endoscopically placed duodenal-jejunal bypass sleeve devices, such as the EndoBarrier (GI Dynamics, Lexington, Massachusetts, USA), have been designed to create an impermeable barrier between chyme exiting the stomach and the mucosa of the duodenum and proximal jejunum. The non-surgical and reversible nature of these devices represents an attractive therapeutic option for patients with obesity and T2DM by potentially improving glycaemic control and reducing their weight.

**Methods and analysis** In this multicentre, randomised, controlled, non-blinded trial, male and female patients aged 18–65 years with a body mass index 30–50 kg/m$^2$ and inadequately controlled T2DM on oral antihyperglycaemic medications (glycosylated haemoglobin (HbA1c) 58–97 mmol/mol) will be randomised in a 1:1 ratio to receive either the EndoBarrier device (n=80) for 12 months or conventional medical therapy, diet and exercise (n=80). The primary outcome measure will be a reduction in HbA1c by 20% at 12 months. Secondary outcome measures will include percentage weight loss, change in cardiovascular risk factors and medications, quality of life, cost, quality-adjusted life years accrued and adverse events. Three additional subgroups will investigate the mechanisms behind the effect of the EndoBarrier device, looking at changes in gut hormones, metabolites, bile acids, microbiome, food hedonics and preferences, taste, brain reward system responses to food, eating and addictive behaviours, body fat content, insulin sensitivity, and intestinal tissue gene expression.

---

### Strengths and limitations of this study

► This study will represent the largest randomised controlled trial of the EndoBarrier device compared with conventional medical therapy, diet and exercise over a treatment period of 1 year and will also provide the longest follow-up data (1 year) of any trial to date.

► This study will provide (1) unique data on the mechanism of action of the duodenal-jejunal bypass sleeve and the effect of foregut exclusion on an individual's metabolic profile, (2) a cost-effectiveness analysis, (3) quality-of-life assessment outcomes and (4) extensive safety data.

► The unblinded design of this trial introduces the risk of bias.

---

**Trial registration number** ISRCTN30845205, ClinicalTrials.gov Identifier NCT02459561.

## BACKGROUND

Recent years have witnessed a global increase in obesity and obesity-related diseases. In 2014, it was estimated that 39% of the world population were overweight (clinically defined as a body mass index (BMI) of 25–30 kg/m$^2$) and 13% were obese (BMI ≥30 kg/m$^2$), and it has been projected that there will be an additional 11 million obese adults in the UK by 2030.[1] Being overweight or obese increases the risk of developing 'metabolic syndrome' and is the main modifiable risk factor for developing insulin resistance

and type 2 diabetes mellitus (T2DM). Having a BMI of >25 kg/m$^2$ increases the risk of developing T2DM by five times, and 90% of adult patients with T2DM are obese or overweight.[2] The prevalence of T2DM has therefore also increased in recent years with an estimated 7.4% of the UK population currently affected and is projected to increase by a further 2.1% in the next 15 years.[3] Compared with the general population, patients with T2DM are 87.6% more likely to be admitted to hospital for a myocardial infarction, 121.1% more likely to be admitted for heart failure, 59.1% for a stroke and are 32% more likely to die prematurely.[4] This represents a significant socioeconomic burden for a largely preventable condition, with combined healthcare costs for these conditions estimated to increase by up to £2 billion each year in the UK.[5]

Adipose tissue is a highly active endocrine organ and acts to modulate metabolism by releasing proinflammatory cytokines (tumour necrosis factor-α, interleukin-6, monocyte chemoattractant protein-1),[6] hormones (leptin and adiponectin), glycerol, and importantly, non-esterified fatty acids.[7–11] In obesity, especially those with centrally placed adipose tissue, there is increased production of many of these mediators that leads to the development of insulin resistance and pancreatic beta cell dysfunction. T2DM occurs when an already insulin-resistant individual develops beta cell dysfunction and is therefore unable to produce the necessary amount of insulin that is required to maintain normoglycaemia, and as a result hyperglycaemia predominates.

Dietary modification, exercise and hypoglycaemic medication remain the mainstay of management for patients with T2DM. Unfortunately, these measures have generally suboptimal and poorly sustained outcomes. Bariatric, or metabolic, surgery remains the most effective long-term means of treating these patients by producing usually profound and sustained weight loss and weight loss independent improvements in insulin secretion and sensitivity, consequently ameliorating, or even eliminating, associated comorbidities and reducing mortality. Roux-en-Y gastric bypass (RYGB) surgery can achieve approximately 23%–35% weight loss, and 72%–90% of patients with T2DM undergoing RYGB are able to achieve sustained euglycaemia without oral hypoglycaemic agents.[12–19] There are several mechanisms by which these outcomes are achieved, namely (1) gastric exclusion from food by producing a small gastric pouch, (2) exclusion of food from the duodenum and proximal jejunum, (3) early delivery of food to the terminal ileum and (4) disrupted bile flow. Within the first few days and weeks following surgery, before weight loss has occurred, early improvements in glycaemic control occur through rapid modulation of hepatic insulin resistance (causing reduced hepatic glucose output). This is then followed by sustained long-term weight loss, via entero-neuro-hormonal mechanisms, with an associated reduction in peripheral insulin resistance.[17 20]

Rubino et al[21] demonstrated in 2006 that the foregut plays a key role in the metabolic changes that occur following bariatric surgery.[21] They demonstrated that exclusion of the proximal small bowel, as occurs in RYGB and similar procedures such as biliopancreatic diversion±duodenal switch, results in improved glucose tolerance that occurs independently of effects from reductions in food intake and body weight, malabsorption or nutrient delivery to the hindgut. These findings have further been substantiated in other studies.[21–24] The proposed mechanisms by which these changes occur include decreased secretion of orexigenic hormones (ghrelin); increased secretion of glucose-dependent insulinotropic polypeptide and cholecystokinin; increased anorexigenic and incretin hormone secretion (eg, glucagon-like peptide 1 (GLP-1), peptide YY (PYY), oxyntomodulin); and increased circulating concentrations of plasma bile acids. Additionally, stimulation of vagal afferent nerves in the small bowel causes entero-neuro-endocrine modulation within the gut–liver–brain axis. The resulting net effects include increased insulin secretion, decreased glucagon secretion, decreased hepatic glucose output, increased pancreatic beta cell mass (via increased proliferation and decreased apoptosis), increased insulin sensitivity, decreased hunger, early satiety and altered food preferences and hedonics, and brain reward system responses away from high-energy foods.[20 25–35]

Such observations have led to the development of novel, endoscopically placed duodenal-jejunal bypass sleeves (DJBS) or liners. These create an impermeable barrier between chyme exiting the stomach and the intestinal mucosa of the duodenum and proximal jejunum, thus preventing absorption within the foregut. The non-surgical and reversible nature of these devices has sparked much interest in recent years due to the prospect of avoiding the associated surgical mortality and morbidity of bariatric procedures (RYGB: 1-year morbidity 14.9%, 30-day mortality 0.5%).[17] First described by Milone et al[36] in animal models in 2006, the effects of DJBS insertion on reducing weight and potentially improving glycaemic regulation, above that of control interventions, have been validated in five randomised controlled trials (RCTs) and numerous observational studies.[30 37–49] In a recent systematic review and meta-analysis by Rohde et al,[50] they concluded that subjects implanted with DJBS achieved an additional 12.6% weight loss compared with sham controls or dietary intervention alone, and a mean greater weight loss of 5.1 kg. In the largest of the RCTs (DJBS+diet n=38 vs diet alone n=39) among the DJBS arm, a significant reduction in glycosylated haemoglobin (HbA1c) of −0.9% was found.[37] This finding however was not seen in the meta-analysis by Rohde et al, where the mean difference in HbA1c reduction of 0.8% was non-significant.[50] Finally, evidence exists for DJBS having positive effects on other metabolic parameters, including blood pressure and serum lipid profile.[37 44 45]

The EndoBarrier DJBS (GI Dynamics, Lexington, Massachusetts, USA) is delivered endoscopically and comprises a nitinol metal anchor, which is used to reversibly affix the device to the wall of the duodenal bulb, and

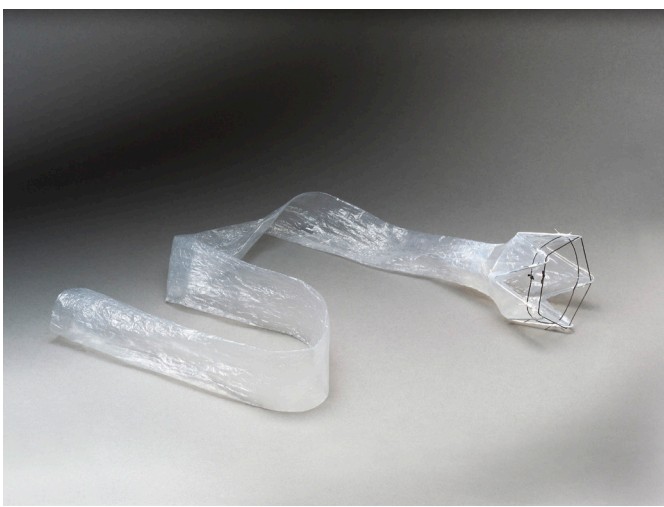

**Figure 1** EndoBarrier gastrointestinal bypass liner.

an impermeable fluoropolymer sleeve that extends 60 cm through the duodenum and into the jejunum (figure 1). The implant is open at both ends to allow for passage of chyme from the stomach into the lower jejunum and prohibits nutrient absorption along its length by creating a barrier between the partially digested food and the absorptive surface of the small intestine. While the chyme passes through the inside of the EndoBarrier device, all bile and pancreatic secretions pass on the outside the liner and only mix with the food when they come into contact at the end of the sleeve.

Robust evidence for the clinical use of the DJBS is hence still lacking. The small number of published trials include small participant numbers with high degrees of intertrial heterogeneity, and the results are therefore not generalisable to routine clinical practice. Mechanistic data is also limited. There is therefore a call for more long-term, high-quality trial data to validate the efficacy and mechanism of action of this device as a potential tool in the treatment of obesity and metabolic syndrome. In this paper, we describe the methodology for a government-funded RCT comparing DJBS against best practice medical therapy for the treatment of patients with obesity and T2DM with inadequate glycaemic control.

## METHODOLOGY
### Aims and objectives
#### Primary objective and endpoint
The primary objective of this study is to evaluate the efficacy of DJBS compared with conventional medical therapy, diet and exercise on glycaemic control. As defined by the International Diabetes Federation (IDF), a substantial improvement in an individual's metabolic state occurs with an improvement in HbA1c by 20%.[51] Our primary endpoint therefore is a reduction in HbA1c by 20% after 12 months of treatment.

#### Secondary objectives and endpoints
The secondary objective of this study is to evaluate the efficacy, acceptability and cost-effectiveness of DJBS compared against conventional medical therapy, diet and exercise. The following are the secondary endpoints:
1. HbA1c of <6%, equivalent to 42 mmol/mol (this infers optimisation of the metabolic state as defined by the IDF)[51]
2. blood pressure <135/85
3. weight loss >15%
4. reduction in dose/number of medications
5. cost of interventions and related health/social care
6. quality-adjusted life years (QALY) accrued (calculated from area under the 5-Level European Quality of Life-5 Dimensions (EQ-5D-5L questionnaire curve)
7. incremental cost per QALY within the trial period and extrapolated through modelling.

Data will also be obtained to investigate the mechanism of action of the EndoBarrier device via changes in the following:
1. gut hormones, bile acids, microbiome and intestinal gene expression
2. food hedonics and preference, taste, appetite, eating behaviour, and brain reward system responses to food evaluation and addictive behaviours using functional MRI (fMRI)
3. total body and tissue-specific insulin sensitivity, and body fat content.

#### Safety objective
The safety of the EndoBarrier DJBS will be evaluated during this trial and the type and frequency of adverse events shall be reported.

### Research approval
This study shall be conducted in full conformity with the 1964 Declaration of Helsinki and all subsequent revisions. All subjects will give informed written consent.

### Study design
This study is a RCT of the EndoBarrier DJBS compared with conventional medical therapy, diet and exercise for the management of subjects with both obesity and T2DM. Over a 2-year period (1 year of treatment and 1 year follow-up), the study will be performed over two investigational sites in the UK: Imperial College Healthcare NHS Trust in London and University Hospital Southampton NHS Foundation Trust. The overall schema for the trial is summarised in figure 2. To ensure that the study is adequately powered and allowing for dropouts, n=80 patients will be randomised into each of the two treatment arms equally across the two sites (table 1).

In order to investigate the mechanism of the effect of the EndoBarrier device, both treatment arms will be divided into three optional subgroups, which will have the following additional assessments during the course of the trial:

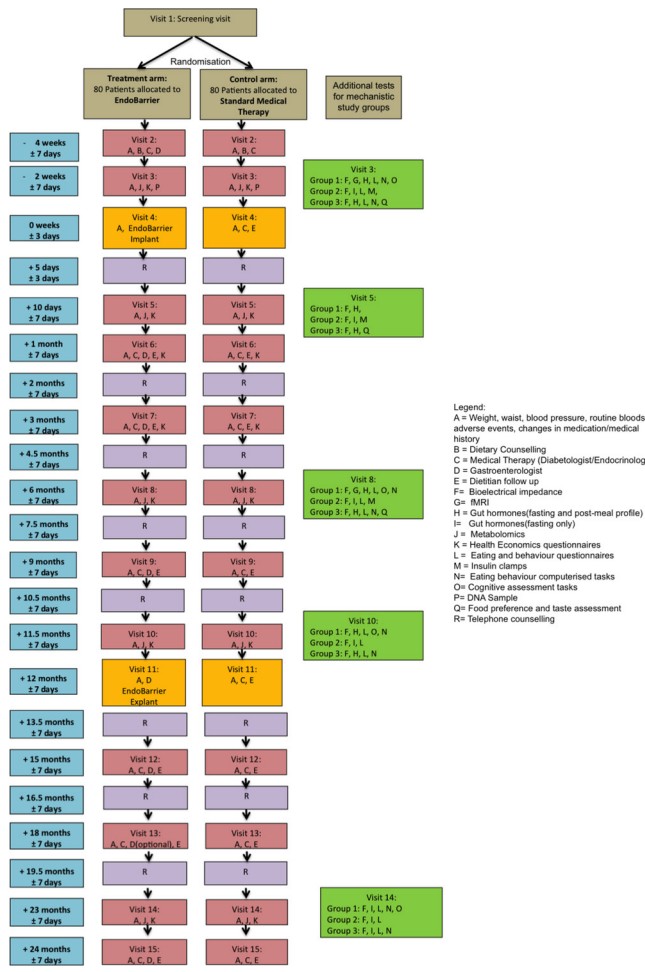

**Figure 2** Study interventions and follow-up schedule.

- ► subgroup 1: fMRI of food reward and addictive behaviours, eating behaviour assessment and postmeal gut hormones
- ► subgroup 2: euglycaemic-hyperinsulinaemic clamps (total body and tissue-specific insulin resistance)
- ► subgroup 3: assessment of taste and food preference, eating behaviour assessment and postmeal gut hormones.

Table 2 summarises the visit schedule, the data to be collected across both study arms and supplementary data that will be collected from the three optional mechanistic subgroups. In addition to routine follow-up visits, all patients will receive regular telephone counselling from

**Table 1** Summary of treatment group

| Treatment group | Subjects (n) | Treatment period 1 (months) | Follow-up period 2 (months) |
| --- | --- | --- | --- |
| EndoBarrier device | 80 | 12 | 12 |
| Standard medical therapy | 80 | 12 | 12 |
| Total number of subjects | 160 | | |

a specialist dietitian to assess their well-being and motivation in the trial.

### Study population
The study population will include male and female patients, aged 18–65 years, with a BMI 30–50 kg/m² and confirmed diagnosis of T2DM for at least 1 year, who have inadequate glycaemic control and are on oral antihyperglycaemic medications (see box for complete inclusion and exclusion criteria).

### Study recruitment
Participants will be identified from several areas across primary, secondary and tertiary healthcare and community settings:
1. diabetes research registers (eg, Diabetes Alliance for Research in England, REC 2002/7/118), hospital or general practice patient databases (participant identification centres), patients referred to diabetes and bariatric specialist clinics, and other research studies within the Imperial College Healthcare NHS Trust and the Local Clinical Research Network
2. study websites; local and national media—websites, radio, newspaper articles and adverts; posters; diabetes, obesity and other support groups; and social media websites.

Potential patients who, after reading a summary Patient Information Sheet (PIS), would like to enter the trial will give their verbal consent for preliminary telephone screening to check basic inclusion and exclusion criteria. Written consent will then be taken from patients to allow the study team to contact their general practitioners (GPs) for the purpose of obtaining additional information on patients' medical history and current medical therapies, and to identify any other clinical reasons as to why patients should not participate. Patients who appear to meet eligibility criteria will be provided with a full trial PIS and then invited to a formal screening visit at one of the study centres. At this stage the patient will be fully informed of the nature of the study and given relevant information about the objectives of the research, benefits and possible adverse events, verbally and in writing. The patient will have the opportunity to ask questions about the trial and formal written consent will be taken for the patient to participate in the main study plus additional consent if they would like to participate in one of the three optional mechanistic subgroups. Once consent has been obtained, the subject's full eligibility will be checked against all inclusion and exclusion criteria (box). Each patient will be informed of his/her eligibility for the trial once all results are available (usually within 1 week from obtaining consent).

### Randomisation
Eligible patients will be randomised into one of the two trial arms using the InForm Integrated Trial Management system, a secure web-based data entry platform. This will be programmed with a randomisation schedule by an independent statistician and protect against bias in the randomisation process as group allocation will be concealed and

**Table 2** Summary of study visit schedule

| Activities | Screening | Baseline | | | Treatment | | | | | | | | | | | | Follow-up | | | | | | |
|---|---|---|---|---|---|---|---|---|---|---|---|---|---|---|---|---|---|---|---|---|---|---|---|
| | V1 | V2 | V3 | V4 | T1 | V5 | V6 | T2 | V7 | T3 | V8 | T4 | V9 | T5 | V10 | V11 | T6 | V12 | T7 | V13 | T8 | V14 | V15 |
| Week/month/day | | −4w ±7d | −2w ±7d | −0w ±3d | +5d ±3d | +10d ±3 | +1m ±7d | +2m ±7d | +3m ±7d | +4.5m ±7d | +6m ±7d | +7.5m ±7d | +9m ±7d | +10.5m ±7d | +11.5m ±7d | +12m ±7d | +13.5m ±7d | +15m ±7d | +16.5m ±7d | +18m ±7d | +19.5m ±7d | +23m ±7d | +24m ±7d |
| Informed consent | X | | | | | | | | | | | | | | | | | | | | | | |
| Inclusion and exclusion criteria | X | | | | | | | | | | | | | | | | | | | | | | |
| Demographics | X | | | | | | | | | | | | | | | | | | | | | | |
| Medical history (including meds) | X | | | | | | | | | | | | | | | | | | | | | | |
| Physical examination | X | | | | | | | | | | | | | | | | | | | | | | |
| ECG | X | | | | | | | | | | | | | | | | | | | | | | |
| Vital signs | X | X | X | X | | X | X | | X | | X | | X | | X | X | | X | | X | | X | X |
| Body weight | X | X | X | X | | X | X | | X | | X | | X | | X | X | | X | | X | | X | X |
| Height | X | | | | | | | | | | | | | | | | | | | | | | |
| Waist circumference | X | X | X | X | | X | X | | X | | X | | X | | X | X | | X | | X | | X | X |
| Routine blood tests | X | X | X | | | X | X | | X | | X | | X | | X | X | | X | | X | | X | X |
| Urine dipstick and female pregnancy test | X | | | | | | | | | | | | | | | | | | | | | | |
| Changes in medical history/medication | | X | X | X | | X | X | | X | | X | | X | | X | X | | X | | X | | X | X |
| Randomisation | | X | | | | | | | | | | | | | | | | | | | | | |
| Health economic questionnaires | | | X | | | | X | | | | X | | | | X | | | | | | | X | |
| Dietary counselling | | X | | C | | | | | | | | | | | | | | | | | | | |
| Dietitian follow-up | | | | | | | X | | | | X | | X | | X | X | | X | | X | | X | X |
| Urine albumin:creatinine ratio | | | | | | X | | | | | | | | | X | | | | | | | X | |
| Reporting of AEs | | X | X | | | X | X | | X | | X | | X | | X | X | | X | | X | | X | X |
| DNA and RNA sampling | | | X | | | X | | | | | X | | | | | X | | | | | | | X |
| Telephone counselling | | | | | X | | | X | | X | | X | | X | | | X | | X | | X | | |
| Diabetologist review | | X | X | C | | | X | | X | | X | | X | | | C | | X | | X | | | X |
| Metabolomics | | | X | | | X | | | | | X | | | | X | | | | | | | X | |

Continued

**Table 2** Continued

| Activities | Screening V1 | Baseline V2 | V3 | V4 | T1 | V5 | V6 | T2 | V7 | T3 | V8 | T4 | V9 | T5 | V10 | V11 | T6 | V12 | T7 | V13 | T8 | V14 | V15 |
|---|---|---|---|---|---|---|---|---|---|---|---|---|---|---|---|---|---|---|---|---|---|---|---|
| | | | | | | | | | | | **Treatment** | | | | | | | | | **Follow-up** | | | |
| Bioelectrical impedance | | | X | | | X | | | | | X | | | | X | | | | | | | X | |
| EndoBarrier group only | | | | | | | | | | | | | | | | | | | | | | | |
| PPI and *Helicobacter pylori* test | | X | | | | | | | | | | | | | | | | | | | | | |
| Distribution of proton pump inhibitors | | T | | | | | | | | | | | | | | | | | | | | | |
| EndoBarrier implant | | | | T | | | | | | | | | | | | | | | | | | | |
| Preparation for EndoBarrier removal | | | | | | | | | | | | | T | | | | | | | | | | |
| EndoBarrier removal | | | | | | | | | | | | | | | | T | | | | | | | |
| Biopsies during implant and explant | | | | T | | | | | | | | | | | | T | | | | | | | |
| Gastroenterologist appointment | | T | | | | | | | | | | | | | | | | | | | | | |
| Subgroups | | | | | | | | | | | | | | | | | | | | | | | |
| Fixed/test meal and postmeal gut hormones and metabolites (groups 1 and 3) | | | X | | | X | | | | | X | | | | X | | | | | | | | |
| Gut hormones and metabolites (fasting only) (groups 1–3) | | | X | | | X | | | | | X | | | | X | | | | | | | X | |
| Food diaries (groups 1–3) | | | X | | | X | | | | | X | | | | X | | | | | | | X | |
| Eating and behaviour questionnaires (groups 1–3) | | | X | | | X | | | | | X | | | | X | | | | | | | X | |
| Appetite Visual Analogue Scales (groups 1–3) | | | X | | | X | | | | | X | | | | X | | | | | | | X | |
| Eating behaviour computerised tasks (groups 1 and 3) | | | X | | | X | | | | | X | | | | X | | | | | | | X | |

Gastroenterologist appointment (T values): T at V2, V6, V7, V9, V11, V12, V13 (T*), V15 (**T**)

Continued

**Table 2** Continued

| Activities | Screening | Baseline | | | Treatment | | | | | | | | | | | | | Follow-up | | | | | |
|---|---|---|---|---|---|---|---|---|---|---|---|---|---|---|---|---|---|---|---|---|---|---|---|
| | V1 | V2 | V3 | V4 | V5 | T1 | V6 | T2 | V7 | T3 | V8 | T4 | V9 | T5 | V10 | V11 | T6 | V12 | T7 | V13 | T8 | V14 | V15 |
| Metal check form (group 1) | X | | | | | | | | | | | | | | | | | | | | | | |
| Handedness questionnaire (group 1) | X | | | | | | | | | | | | | | | | | | | | | | |
| Additional pregnancy tests | | | F | | | | | | | | F | | | | | | | | | | | | |
| DS-R disgust questionnaire (group 1) | | | X | | | | | | | | | | | | | | | | | | | | |
| Functional MRI (group 1) | | | X | | | | | | | | X | | | | | | | | | | | | |
| Insulin clamps (group 2) | | | X | | X | | | | | | X | | | | | | | | | | | | |
| Cognitive assessment tasks (group 1) | | | X | | | | | | | | X | | | | X | | | | | | | X | |
| Food preference/ taste assessment (group 3) | | | X | | X | | | | | | X | | | | | | | | | | | | |
| 24-Hour dietary recall (group 3) | | | X | | X | | | | | | X | | | | X | | | | | | | X | |

*Optional (at request of the patient).

X, performed in all patients unless otherwise stated; F, performed in females only; C, performed in control arm (standard medical therapy) only; T, performed in treatment arm (EndoBarrier) only.

PPI, proton pump inhibitor; AE, adverse event; DS-R, disgust scale-revised.

## Box Study inclusion and exclusion criteria

**Inclusion criteria**
► Age 18–65 years (male or female)
► Type 2 diabetes mellitus for at least 1 year
► HbA1c 7.7%–11.0% equivalent to 58–97 mmol/mol
► On oral hypoglycaemic medications
► BMI 30–50 kg/m$^2$

**Exclusion criteria**
► Language barrier, mental incapacity, unwillingness or inability to understand and be able to complete questionnaires
► Non-compliance with eligibility criteria
► Females of childbearing potential who are pregnant, breast feeding, or intend to become pregnant or are not using adequate or reliable contraceptive methods
► Evidence of absolute insulin deficiency as indicated by clinical assessment, a long duration of type 2 diabetes mellitus and a fasting plasma C-peptide of <333 pmol/L
► Current use of insulin
► Previous diagnosis with type 1 diabetes mellitus or a history of ketoacidosis
► Requirement of non-steroidal anti-inflammatory drugs or prescription of anticoagulation therapy during the implant period
► Current iron deficiency and/or iron deficiency anaemia
► Symptomatic gallstones or kidney stones at the time of screening
► History of coagulopathy, upper gastrointestinal bleeding conditions such as oesophageal or gastric varices, congenital or acquired intestinal telangiectasia
► Previous gastrointestinal surgery that could affect the ability to place the device or the function of the implant
► History or presence of active *Helicobacter pylori* (if subjects are randomised into the EndoBarrier arm and have a history or presence of active *H. pylori* tested at study visit 2, they can receive appropriate treatment and then subsequently enrolled into the study)
► Family history of a known diagnosis or pre-existing symptoms of systemic lupus erythematosus, scleroderma or other autoimmune connective tissue disorder
► Severe liver impairment (ie, AST, ALT or gGT >4 times upper limit of the reference range) or kidney impairment (ie, estimated glomerular filtration rate <45 mL/min/1.73 m$^2$)
► Severe depression, unstable emotional or psychological characteristics (including Beck Depression Inventory II score>28)
► Poor dentition and inability to adequately chew food
► Planned holidays up to 3 months following the EndoBarrier implant

ALT, alanine transaminase; AST, aspartate transaminase; BMI, body mass index; gGT, gamma-glutamyl transpeptidase; HbA1c, glycosylated haemoglobin.

automatic. The randomisation will be at a ratio of 1:1 and stratified by site and two BMI groups, 30–40 and 40–50 kg/m$^2$. Each patient will be informed of his/her randomisation allocation and will be assigned a unique study identification number. Only the subject number and initials will be recorded in the case report form (CRF). All other patient-identifiable data will be completely anonymised.

### Trial interventions
#### EndoBarrier gastrointestinal liner
The EndoBarrier gastrointestinal liner device received CE mark for 12 months' implant duration on 11 December 2009 and is a single-use, minimally invasive device used to achieve weight loss and improve T2DM status in subjects who are obese (figure 1).

At visit 2 (–4 weeks), participants who have been randomised to receive the EndoBarrier device will be tested for the presence of *Helicobacter pylori*, either by faecal antigen or urea breath testing. Those patients testing positive will be offered 1 week of triple-eradication therapy, as per guidance published within the British National Formulary (BNF), and will then be retested after a further 4 weeks to confirm complete eradication before continuing with implantation of the EndoBarrier device. Subsequently, all patients will be prescribed a proton pump inhibitor (omeprazole 40 mg twice daily) and instructed to commence this 3 days prior to the implant procedure. They will continue this for the duration of the implant period (12 months) and for a further 2 weeks following device removal.

At visit 4 (0 weeks), after an 8-hour fast, subjects will have the EndoBarrier device implanted under general anaesthesia. The implant is delivered endoscopically on a custom catheter and the anchor is sited in the duodenal bulb using a custom delivery system under fluoroscopic X-ray guidance (mean fluoroscopic X-ray time for insertion is 7 min, range 1–20 min). The 60 cm sleeve is unfurled and then the final positioning plus patency is confirmed by assessing for the free flow of radio-opaque contrast through the device. Videos and photos of the fluoroscopy images are recorded to help the investigators make treatment decisions. During implantation eight gastric and small bowel biopsies will be taken using standard biopsy forceps. Four biopsies will be used for routine histology and four biopsies will be used for RNA extraction to perform genome-wide expression analysis. Participants will be discharged from hospital the same day with an implant information card, which describes the implant, identifies who to call in case of an emergency and what symptoms to look for following the implant. Subjects will have their dose of sulfonylurea medication reduced by 50% at the time of EndoBarrier implant to avoid potential hypoglycaemic episodes.

The device will be removed at visit 11 (after 12 months) under sedation or general anaesthesia. The gastroscope, which is fitted with a foreign body retrieval hood, is used to locate the implant and a custom grasper is passed through the working channel of the gastroscope to grab a polypropylene tether located on the proximal portion of the anchor. Pulling on this tether will collapse the proximal end of the anchor, which can then be pulled into the foreign body hood and removed by withdrawing the gastroscope through the subject's mouth. During this removal, eight further biopsies will be taken for histology and RNA extraction. Following removal of the EndoBarrier device, patients will be followed up for a further 12 months.

### Diabetes review
Participants in both arms of the trial will have their T2DM managed in accordance with the guidelines of

the American Diabetes Association (ADA).[52 53] These guidelines have been chosen as they would adhere to the current best worldwide practice that would still be relevant when the results are published following study completion. Both treatment groups will have a review of their T2DM by a suitably trained physician at visits 2, 6, 7, 9, 12, 13 and 15. Additionally, the standard care arm of the trial will have an additional review at visits 4 and 11 in place of the EndoBarrier implant and removal. Adjustments to a patient's oral antihyperglycaemic medication and escalation of therapy are at the investigators' discretion and will comply with general recommendations laid out by the ADA.[53]

### Dietary counselling and physical activity

At visit 2, all patients' historical and current eating behaviours will be assessed by a qualified dietitian using the following information: anthropometry; biochemistry; comorbidities; activity levels; eating habits including previous diets; lifestyle including smoking, drug and alcohol misuse; weight history; psychiatric history; family history of obesity, diabetes, mental illness or eating disorders; available support network; work status; and readiness and motivation for change. Patients will then receive dietary and physical activity counselling in accordance with local standards with the intention of providing each subject with lifestyle/behavioural modification information and good eating practices. In addition, subjects in the EndoBarrier arm will receive written information on how their diet will change after implantation of the device and they will receive specialist guidance for eating with their EndoBarrier.

All patients will be reviewed by a specialist dietitian at visits 2, 6, 7, 9, 12, 13 and 15. In addition, participants in the standard care arm of the trial will have an additional review at visits 4 and 11 in place of the EndoBarrier implant and removal. During the course of the trial, participants will be recommended to consume 600 kcal less every day, depending on their age, gender, activity levels and body weight. Guidelines for daily amounts are between 1200 and 1500 kcal for women and between 1500 and 1800 kcal for men. In accordance with standard dietary practice, subjects will be advised to eat regularly every day (five times per day); to control their portion sizes and intake of carbohydrates/starchy foods; to increase their intake of low glycaemic index and high-protein foods, as well as vegetables; and to reduce their intake of foods high in fat and sugar, and alcohol. Participants will be advised to include more physical activity in their daily routine and encouraged to do more activity in their leisure time. Their goal will be to include 150 min a week of moderate intensity and 75 min a week of vigorous intensity aerobic activity and muscle-strengthening activities, on more than 2 days a week. Changes in physical activity level will be monitored using the International Physical Activity Questionnaire.[54]

### Liquid diet

To avoid disruption of the device in the immediate period following implantation, patients will follow a liquid diet for the 7 days before and 13 days (±3 days) after the intervention visit (visit 4). The liquid diet will be guided by the specialist dietitian and will comprise 125 mL Fortisip Compact drinks (Nutricia, Trowbridge, UK): five per day for men, four per day for women, containing per 100 mL 240 kcal, 9.6 g protein (16% total energy), 29.7 g carbohydrate (49%), 15 g sugars and 9.3 g fat (35%). Patients will also be allowed to consume sugar-free squashes, smooth/clear soup (one medium bowl per day), tea or coffee without sugar, or unsweetened puree. To standardise both therapy groups, all patients across both arms will follow the liquid diet for this duration and period of the study.

### Assessment of objectives

#### Assessment of primary objective

Each study participant will have their International Federation of Clinical Chemistry HbA1c measured at screening and then subsequently at visits 5, 7, 8, 9, 10, 12, 13 and 15. Samples will be processed at the laboratory local to each study centre using standard methods. Results will be recorded on the InForm system.

#### Assessment of secondary objectives

Individuals in both study arms will be invited for regular medical check-ups (figure 2), which will include routine anthropometric measurements (height, weight, waist circumference, pulse and blood pressure) and blood tests (table 3). Any changes to the participants' health or medications will be carefully documented on the CRF, and all adverse events will be reported in detail in line with the standard principles of Good Clinical Practice.

Patients in both treatment arms will be asked to complete health economics questionnaires at visits 3, 5, 6, 7, 8, 10 and 14. These comprise the EQ-5D-5L questionnaire to assess health-related quality of life and a bespoke questionnaire designed to collect information about patients' use of health and social care resources (for costing purposes).[55] The Resource Use Questionnaire will be adapted from existing instruments and will include[56 57] the following:

1. medications for diabetes, weight loss, blood pressure, lipid control and cardiovascular disease;
2. primary care consultations (with GP, nurse or other healthcare professional); hospital outpatient clinic visits (by specialty); and Emergency Department attendances (admitted/not-admitted)
3. inpatient stays and procedures, investigations, and use of any other National Health Service (NHS)-related community health and social services (eg, chiropody).

Costs for private health and social care, out-of-pocket expenditure by patients and 'indirect costs' per patient time will not be included.

In addition, data will be collected in the CRF from hospital information systems and case notes to cost the EndoBarrier intervention and the diet/exercise

**Table 3** Summary of blood tests at each study visit

| Blood test | V1 | V3 | V5 | V6 | V7 | V8 | V9 | V10 | V11 | V12 | V13 | V14 | V15 |
|---|---|---|---|---|---|---|---|---|---|---|---|---|---|
| Haematology (full blood count) | x | x | x | x | x | x | x | x | x | x | x | x | x |
| Routine biochemistry (including urea and electrolytes) | x | x | x | x | x | x | x | x | x | x | x | x | x |
| Liver function tests | x | x | x | x | x | x | x | x | x | x | x | x | x |
| Fasting glucose | x | x | x | x | x | x | x | x | x | x | x | x | x |
| Creatinine | x | x | x | x | x | x | x | x | x | x | x | x | x |
| HbA1c | x |  | x |  | x | x | x | x |  | x | x |  | x |
| Fasting lipids (cholesterol, HDL, LDL, triglycerides) | x | x | x | x | x | x | x | x |  | x | x | x | x |
| C-peptide | x |  |  |  |  |  |  |  |  |  |  |  |  |
| Insulin (fasting) | x | x | x |  |  | x |  | x |  |  |  | x |  |
| Vitamin D | x |  |  |  |  | x |  | x |  |  |  |  |  |
| Iron studies | x |  |  |  |  | x |  | x |  |  |  |  |  |
| Vitamin B$_{12}$ | x |  |  |  |  | x |  | x |  |  |  |  |  |
| Serum folate | x |  |  |  |  | x |  | x |  |  |  |  |  |
| Free thyroxine | x |  |  |  |  |  |  | x |  |  |  |  | x |
| TSH | x |  |  |  |  |  |  | x |  |  |  |  | x |
| Cortisol (subgroup 1 only) |  | x |  |  |  | x |  |  |  |  |  |  |  |
| Oestradiol (subgroup 1 only) |  | x |  |  |  | x |  |  |  |  |  |  |  |
| Progesterone (subgroup 1 only) |  | x |  |  |  | x |  |  |  |  |  |  |  |
| LH (subgroup 1 only) |  | x |  |  |  | x |  |  |  |  |  |  |  |
| FSH (subgroup 1 only) |  | x |  |  |  | x |  |  |  |  |  |  |  |

FSH, follicle-stimulating hormone; HbA1c, glycosylated haemoglobin; HDL, high-density lipoprotein; LDL, low-density lipoprotein; LH, luteinising hormone; TSH, thyroid-stimulating hormone.

intervention. Information to be collected will include the following:

1. routine assessments required before implantation and removal of the EndoBarrier; disposables and staff time for the insertion and removal procedures, including day case and overnight stay if required; treatment of any adverse events related to the procedures (additional inpatient stays, clinic visits and readmissions); and dietitian time to deliver the diet and physical activity counselling and for telephone follow-up
2. routine hospital follow-up and diabetes care and hospital treatment for cardiovascular events or other complications of diabetes.

## Mechanistic study groups

In addition to the routine data collected above, at visits 3, 5, 8, 10 and 14, mechanistic data will be gathered from all patients across both study arms:

1. body fat mass (kg and % of body weight) measured by bioelectrical impedance analysis
2. collection of stool, urine and plasma for assessment of metabonomics using nuclear magnetic resonance spectroscopy and mass spectrometry, and microbiome analysis; measurement of insulin, gut hormones (ghrelin, GLP-1, PYY), bile acids, leptin and other ad- ipocytokines, and markers of insulin resistance and inflammation from venous blood samples (fasting for all visits and then following a meal in subgroups 1 and 3 at visits 3, 5, 8 and 10); DNA and RNA from venous blood samples for examination of genetic variants that may predict weight loss, cause or contribute to obesity; and urinary albumin:creatinine ratio
3. comprehensive 3 day food diary

### Subgroup 1: fMRI

Subjects in each study arm, at the London site only, will have fMRI scans to examine brain function related to food reward and addictive behaviours contributing to overeating at baseline (visit 3) and at 6 months (visit 8) after intervention. These will be supplemented by and correlated with psychological questionnaires, computerised tasks and test meals at these and other visits. The study visits will last up to 6–8 hours (scanning visits 3 and 8). Subjects will have structural and fMRI brain scans lasting up to 90 min using a 3.0 Tesla Siemens Verio MR scanner after an overnight fast. While in the scanner, subjects view a mirror reflecting a computer screen and can respond to instructions using a keypad held in their hand.

The following anatomical brain scans will be collected at both visits:

1. anatomical T1-weighted and T2-weighted MRI scans to provide structural neuroimaging data and allow image registration to standard space
2. diffusion tensor imaging to examine white matter tract integrity.

The following resting state fMRI scans will be collected at both visits:

1. arterial spin labelling to measure resting regional cerebral blood flow[58 59]
2. resting state blood oxygen level-dependent (BOLD) fMRI to measure resting-state functional connectivity.[60]

The following task-related fMRI scans will be collected at both visits:

1. food picture evaluation task: to assess reward system activation when subjects view a variety of different pictures (high-energy and low-energy foods, household objects, blurred pictures as a baseline) and simultaneously rate how 'appealing' the pictures are using the keypad as a measure of anticipatory food reward or food due reactivity[32 34 61–63]
2. monetary incentive delay task: a game in which subjects need to press a button during a specific time window when given a cue on the computer screen in order to win or prevent the loss of hypothetical monetary prizes to assess anticipatory non-food reward responsivity[64–66]
3. Go-No-Go task: to assess motor response inhibitory control as a measure of compulsivity[67–69]; the task contrasts brain activation during responses to infrequent no-go signals (eg, 'do not press' button when viewing one symbol) compared with an implicit go baseline (eg, 'do press' button when viewing a different symbol)[66]
4. negative emotional reactivity task: to assess brain response during viewing of unpleasant, negatively valent pictures compared with neutral control pictures.[66]

Study visits without scanning at visits 5, 10 and 14 will last 3–6 hours. At visits 3, 8, 10 and/or 14, subjects in subgroup 1 will also complete several questionnaires and perform several computer-based tasks to assess eating and addictive behaviours and cognition, including the following:

1. Wechsler Test of Adult Reading (WTAR) : to document baseline intellectual status (visit 3 only)
2. Kirby Delay Discounting Task: to assess temporal impulsivity to a hypothetical non-food monetary rewards (visits 3, 8, 10 and 14)[70]
3. Leeds Food Preference Questionnaire: to assess bias of food preference to foods high in fat and sugar and explicit and implicit liking of foods high/low in fat/ sugar (visits 3, 8, 10 and 14)[71]
4. Progressive Ratio Task (PRT): to measure breakpoint of effort that subjects are willing to spend by having to press a computer mouse an increasing number of times to receive a chocolate M&M sweet, to assess appetitive food reward and motivation (visits 3 and 8).[33 34]

An ad libitum test meal will be performed at visits 3 and 8, in which subjects first taste and rate the palatability, intensity and acceptability of different foods high or low in fat and sugar, and then eat as much of whichever foods they want, to assess food taste, preference and choice, total energy intake and macronutrient composition. Subjects will also complete visual analogue ratings of appetite, anxiety, stress and sleepiness, and measurement of fasting and postmeal hormones and metabolites over the study visit.[32 63]

At visit 5, 10 and 14 subjects will consume a fixed mixed meal tolerance test with measurement of fasting and postprandial hormones and metabolites.

### Subgroup 2: euglycaemic-hyperinsulinaemic clamp

On visits 3, 5 and 8, patients in each study arm, at the Southampton site only, will undergo a euglycaemic-hyperinsulinaemic clamp with stable isotope infusion to determine overall insulin and compartment-specific insulin sensitivity (liver, muscle and adipose depot). Patients will be instructed to consume a standardised meal or meal replacement the evening prior to their study visit. A venous catheter will be inserted into a vein of each arm on the study morning. The first cannula will be used for infusions and the other for blood sampling. If blood glucose levels are higher than 6 mmol/L on arrival, then a variable rate insulin infusion will be started to attain a stable glucose level (4.0–6.0 mmol/L) prior to commencement of the euglycaemic-hyperinsulinaemic clamp.

A primed continuous infusion of 6, $6-^2H_2$-glucose, a stable isotope tracer, will be started and maintained for 7 hours. Two hours later a two-stage euglycaemic-hyperinsulinaemic clamp procedure will be started and continued for 5 hours. During stage 1 of the clamp procedure, in which hepatic insulin resistance is assessed, insulin will be infused at a low dose (0.3–0.5 mU/kg/min) for 2 hours. During stage 2 of the clamp procedure, in which peripheral insulin resistance is assessed, insulin will be increased to a higher dose (1.5 mU/kg/min) for 2 hours. Euglycaemia will be maintained by infusing 20% dextrose at a variable rate. Blood samples will be taken every 5 min to measure blood glucose concentration and the dextrose infusion will be adjusted accordingly. The exogenous glucose infusion will be enriched with 6, $6-^2H_2$-glucose to prevent a fall in plasma tracer enrichment and underestimation of endogenous glucose production rate. Blood samples will be obtained before the start of the tracer infusions, every 10 min during the final 30 min of the basal period and stages 1 and 2 of the clamp procedure, and every 30 min between these periods to determine glucose enrichment and concentration, free fatty acid, insulin, C-peptide, glucagon, gut hormones and metabolite concentrations. At the same time points participants will be asked to complete appetite Visual Analogue Scale (VAS).

The isotopic enrichment of plasma glucose will be determined by gas chromatography mass spectrometry at

the Wolfson Centre for Translational Research, Postgraduate Medical School, University of Surrey, UK.

### Subgroup 3: taste and food preference assessment

On visits 3, 5, 8, 10 and 14, patients in each study arm, at both the London and Southampton sites, will attend the research facility after an overnight fast. The total duration of these visits will be up to 7 hours (visits 3, 5, 8 and 10) and 5 hours (visit 14). On the morning of those visits, patients will perform two behavioural computerised tasks: the Kirby Delay Discounting Task to assess temporal impulsivity to non-food monetary reward (visits 3, 8, 10 and 14)[70]; and the Leeds Food Preference Questionnaire to assess bias of food preference to foods high in fat and sugar, and explicit and implicit liking of foods high/low in fat/sugar (visits 3, 8, 10 and 14).[71]

Sweet taste detection testing will be performed at visits 3, 5 and 8 by following the method of constant stimuli in which seven ascending sucrose concentrations in solution will be used to determine sweet detection thresholds.[72] At the same visits consummatory taste reward will be assessed in which five ascending sucrose solutions will be used to test responses in intensity ratings and hedonic reward. To assess the appetitive behaviour towards a sweet and fatty food reward, a PRT will be performed 2 hours after the consumption of a fixed meal.[33]

Total caloric intake and macronutrient composition will be assessed using 3-day food diaries and a 24-hour recall that is carried out by a trained dietitian/nutritionist on all visits. Patients will also complete the European Prospective Investigation into Cancer and Nutrition (EPIC) Food Frequency questionnaire at visits 3, 8 and 10. Finally, a fixed mixed meal tolerance test with measurement of postmeal hormones and metabolites will be performed.

### Additional subgroup-specific procedures and measurements

Across all three subgroups only, the following additional data will also be collected during the mechanistic study visits:

1. trait, state and symptom questionnaires: to evaluate aspects of eating behaviour (eg, dietary restraint, emotional eating, disinhibition, hunger, external eating), reward sensitivity, mood, impulsivity, aversive symptoms, symptoms of dumping syndrome, including Dutch Eating Behavior Questionnaire, Three-Factor Eating Questionnaire, Yale Food Addiction Scale, Binge Eating Scale, Beck Depression Inventory II, Hospital Anxiety and Depression Scale, Barratt Impulsiveness Scale and urgency, premeditation, perseverance, sensation seeking, and positive urgency (UPPS-P) Impulsive Behavior Scale
2. VAS ratings: to assess subjective feelings of hunger, nausea, fullness, sleepiness, stress and anxiety when fasted and during meal tests.

### Power calculation

Conservatively, it was estimated that 15% of patients in the control arm will achieve the target, but we believe this to be an overestimate. The Steno study is the best quality randomised study (n=80 patients in each arm) into the effect of best medical therapy published to date and demonstrated over an average 7.8 years significant improvements in HbA1c among those having intensive medical therapy from 8.4±1.6 to 7.7±1.2, but no change in HbA1c among those continuing with standard medical therapy.[73] This study defines the very best that could realistically be achieved in the control arm, but expect there to be very little if any change in this group. The reporting of HbA1c as an outcome measure was not in accordance with the newly defined IDF criteria, but considering the small average reduction achieved in the Steno study, it will be assumed that a target of 15% of patients reaching the endpoint is a conservative estimate. Company data on the small number of patients who have reached a year with the device in place suggest that 40% will achieve this target.

According to our own experience with the device in a commercially sponsored study, up to 30% of patients in the treatment group may have the device removed early. Nevertheless other commercially sponsored studies of this device have achieved lower explant rates (J Tetreault, GI Dynamics, 2014 - unpublished). To allow for up to 30% early removal, we have therefore diluted the treatment effect from 40% vs 15% to 35% vs 15%, achieving the target of 20% reduction in HbA1c for treatment arm versus standard arm. With these assumptions, n=73 patients per group will give 80% power with a two-sided alpha 0.05 to detect a significant effect. Adding 10% loss of follow-up increases the sample size to n=80 per group.

The dilution was calculated starting from the assumption that 40% of patients with the device will reach the target (this estimate is based on company data based on patients with diabetes in the same range of BMI as in the present proposal). If 30% of patients in the treatment group need to remove the device early but remain available for follow-up, in the worst case scenario, the proportion reaching the target is the same as in the control group, bringing the estimate for the treatment group to 32.5%. However most of them will keep the device for some time, having some benefit, so it is plausible to assume that the estimate is higher than 32.5%. Dividing the main effect 15% vs 40% in three parts, we assume that in the 30% of patients with removal, for one third the same effect will be achieved as in the control group (15% reaching the target), for one third it will be increased (23% reach the target) and for one third more increased (31% reach the target). Overall, this would give an estimate of 35% for the treatment group.

### Statistical analysis

Patient characteristics will be summarised. Summaries of continuous variables will be presented as means and SDs if normally distributed, and as medians and IQRs for skewed data, while categorical variables will be presented as frequencies and percentages. The difference between the two study groups in the proportion of patients

achieving substantial improvement in the metabolic syndrome both at 12 and 18 months will be analysed using logistic regression adjusting for the stratification variables (BMI groups and sites). Analysis of secondary outcomes will be conducted using standard statistical procedures applicable to categorical or continuous data as appropriate. For missing values we will explore the pattern and the extent of missingness, and we will carry out an appropriate form of multiple imputation if required. The analysis will be performed according to the intention-to-treat principle. All statistical tests will be two-tailed with a 5% significance level.

## Metabonomics and microbiome analyses

Metabonomic data sets will be analysed using principal component analysis and orthogonal partial least-squares analysis (O-PLS). The metabolic and microbial data will also be analysed in relation to response measurements such as BMI, gut hormone levels and so on using O-PLS regression analysis and Bayesian approaches. A range of statistical methods will be optimised and applied to the data to identify weight loss and T2DM-associated microbiota and metabolites.

## Health economics

The economic health analysis will be conducted following the National Institute for Health and Care Excellence Reference Case, which includes the use of QALYs as the measure of health outcome, and adoption of an NHS and personal social services perspective for costs (Guide to the methods of technology appraisal 2013, http://publications.nice.org.uk/pmg9).

## Within-trial analysis

EQ-5D-5L health states will be scored using the English value set to give utility values at each time point (−2 weeks, 10 days, 1, 3, 6, 11.5 and 23 months).[74] QALYs will be estimated for each patient using an area-under-the curve approach.

The cost of the EndoBarrier intervention, the lifestyle intervention, and other related health and social care will be estimated from resource use data. Unit costs for the included services will be obtained from standard national sources (BNF or Drug Tariff for drug prices, Department of Health Reference Costs for investigations, procedures and outpatient visits, Personal Social Services Research Unit (PSSRU) estimates for other primary and community health and social services). Total costs will be estimated for each patient over the 24-month trial period.

Patient-level cost and QALY estimates will be combined to estimate an incremental cost-effectiveness ratio for the EndoBarrier device compared with standard medical therapy over the 24-month trial period. The analysis will combine multiple imputation to account for missing EQ-5D and resource use data with bootstrap regression to estimate mean cost and QALYs for the two patient groups. Missing data are often a particular problem for economic analysis, even in studies with good follow-up of primary endpoints, as area-under-the-curve approach requires data from multiple time points. A bootstrap regression approach will be used to account for non-normal distributions of cost data, to adjust for baseline differences in utility or other patient characteristics, and to allow for correlations between costs and QALYs.

## Cost-effectiveness modelling

A decision analytical model will be developed to estimate clinical outcomes, QALYs and costs beyond 24 months. The time horizon for the modelling will be for the remainder of the patients' predicted lifetimes, as recommended in modelling guidelines.[75 76] Before commencing this modelling exercise, a review of published economic decision models for weight loss interventions for people with T2DM will be conducted in order to identify possible model structures and sources of input parameters. The conceptual design of the model will be discussed and agreed among the research team before programming commences. It is anticipated that the model will take the form of an individual patient simulation, using either a discrete time or discrete event approach to simulate the onset of diabetes/obesity-related complications, and hence QALYs and costs under alternative treatment strategies.

Data from the trial will be used to provide estimates of the effect of the EndoBarrier compared with conventional management. In addition, published systematic reviews and meta-analyses of bariatric surgery for similar patient population will be reviewed. If recent evidence of sufficient relevance and quality is available, we will extend our model to include indirect comparisons with these other interventions. Other model parameters will be sourced from targeted literature reviews and routine data sources. The choice of software for the model will be made after specification of the conceptual design. Before use, the model will be validated by an experienced health economist not involved in the development of the model. This will be done using a checklist developed by the Brunel Health Economics Research Group, which includes a range of suggestions for checking that a model is free from errors (verification) and that it is consistent with internal and external data (validation).

Probabilistic sensitivity analysis will be used to estimate the impact of uncertainty over model parameters, and value of information analysis to estimate the value of conducting further research. In addition, deterministic sensitivity analysis will be used to examine the impact of uncertainties over the model structure.

## Gut hormones, metabolites and bile acids

These will be measured in the fasted and/or postprandial state for each patient and compared within and between the groups using parametric/non-parametric repeated-measures statistical testing.

## Food hedonics and brain reward responses

Brain activation during fMRI paradigms and outcomes from behavioural measures of eating and addictive

**Table 4** Serious adverse events from the EndoBarrier device (GI Dynamics safety reporting 2008 to March 2017)

| | 2008 | 2009 | 2010 | 2011 | 2012 | 2013 | 2014 | 2015 | 2016 | 2017 (January –March) | Total |
|---|---|---|---|---|---|---|---|---|---|---|---|
| Distributed devices | 25 | 143 | 157 | 275 | 391 | 812 | 987 | 482 | 383 | 43 | |
| Hepatic abscess | 0 | 0 | 0 | 0 | 1 | 8 | 12 | 9 | 4 | 2 | 36 |
| Hepatic abscess rate | | | | | | | | | | | 1.0% |
| Hepatic abscess with explant ≤12 months | 0 | 0 | 0 | 0 | 1 | 5 | 9 | 7 | 1 | 2 | 25 |
| Hepatic abscess rate | | | | | | | | | | | 0.7% |
| Intolerance | 0 | 5 | 4 | 12 | 5 | 10 | 11 | 4 | 18 | 0 | 69 |
| Intolerance rate | | | | | | | | | | | 1.9% |
| Liner obstruction | 0 | 4 | 0 | 5 | 3 | 1 | 10 | 2 | 0 | 0 | 25 |
| Liner obstruction rate | | | | | | | | | | | 0.7% |
| GI bleed | 0 | 1 | 3 | 6 | 5 | 9 | 20 | 8 | 4 | 1 | 57 |
| GI bleed rate | | | | | | | | | | | 1.5% |
| Migration/Movement | 0 | 10 | 6 | 4 | 1 | 6 | 11 | 13 | 0 | 1 | 52 |
| Migration/Movement rate | | | | | | | | | | | 1.4% |
| Pancreatitis | 0 | 0 | 0 | 0 | 4 | 1 | 7 | 0 | 0 | 0 | 12 |
| Pancreatitis rate | | | | | | | | | | | 0.3% |
| Perforation | 0 | 0 | 1 | 0 | 0 | 5 | 2 | 2 | 1 | 0 | 11 |
| Perforation rate | | | | | | | | | | | 0.3% |
| Surgical removal | 0 | 0 | 0 | 1 | 1 | 0 | 8 | 1 | 1 | 0 | 12 |
| Surgical removal rate | | | | | | | | | | | 0.3% |
| **Total incidence** | **0** | **20** | **14** | **28** | **20** | **40** | **81** | **39** | **28** | **4** | **274** |
| **Total cumulative rate** | | | | | | | | | | | **7.4%** |

behaviours and questionnaires will be compared between groups using a 2×2 analysis of variance design including group (control vs EndoBarrier) as a between-subject factor, time (baseline vs follow-up visit) as a within-subject factor, and group × time interaction to identify differential effects between groups. For fMRI studies, analysis will use regions of interest (eg, for food picture evaluation task: orbitofrontal cortex, amygdala, caudate, nucleus accumbens, anterior insula) and whole brain analyses to compare groups using statistical thresholds of voxel-wise correction false discovery rate $p<0.05$ or cluster-wise family-wise error correction $p<0.05$. Correlations of BOLD signal will be made with other behavioural variables by linear regression analysis to examine the relevance of changes in brain activation.

### Food preference and sweet taste
Dietary energy intake, macronutrient composition, sweet taste detection thresholds and visual analogue taste ratings will be quantified for each patient and compared within and between the groups at different time points using parametric/non-parametric repeated-measures statistical testing. Regressions will be performed with clinical outcomes (eg, weight loss, HbA1c) to identify predictive markers and generate mechanistic hypotheses.

### Euglycaemic-hyperinsulinaemic clamps
Overall and tissue specific insulin sensitivity will be quantified for each patient and compared within and between the groups at three time points using parametric/non-parametric repeated-measures statistical testing.

In addition, linear regression will be performed to correlate mechanistic variables collected from each of the subgroups 1–3 at baseline or during the intervention with clinical outcomes at 1 year, for example, weight loss and decreases in HbA1c, to generate predictive markers and generate mechanistic hypotheses.

### DISCUSSION
Experience of 3717 EndoBarrier devices distributed worldwide has demonstrated a favourable risk-to-benefit ratio (GI Dynamics, February 2017), and their minimally invasive and reversible nature represents a very attractive treatment modality for patients with obesity and T2DM. Evidence already exists in the literature in support of the efficacy of DJBS by reducing weight and potentially improving glycaemic control.[37 38 40 42 44–46 50] Nonetheless, it is reported that up to 100% of patients will experience a non-serious adverse event (SAE) (predominantly abdominal discomfort and nausea immediately following

implantation),[50] and 7.4% will suffer an SAE (GI Dynamics safety reporting 2008 to March 2017). The exact nature of these events is summarised in table 4.

Notably, the pivotal US ENDO trial (EndoBarrier vs sham procedure) was terminated in July 2015 after only 325 subjects were randomised (n=216 EndoBarrier subjects) due to a higher than expected hepatic abscess (HA) rate of 3.5% (compared with a global incidence of 0.73%). This high incidence of HA is not the experience within Europe, with 1.2% being reported in 1901 distributed devices (UK HA rate in 523 cases is 1.34%). There have also been no deaths attributed to the EndoBarrier, and all patients experiencing an SAE have recovered without long-term sequelae.

Research to date therefore validates the EndoBarrier DJBS as a potential treatment option for patients who are obese with or without T2DM. These studies however have been limited by their low participant numbers, short follow-up duration and wide intertrial heterogeneity. Thus, there is a call for more robust clinical trial data to investigate its efficacy, safety and acceptability, and to establish where its use may fall within the treatment algorithm of such patients. This study will represent the largest RCT of the EndoBarrier device compared with conventional medical therapy, diet and exercise over a treatment period of 1 year and will also provide the longest follow-up data (1 year) of any trial to date. Additionally, this study will provide (1) unique data on the mechanism of action of the DJBS and the effect of foregut exclusion on an individual's metabolic profile, (2) a cost-effectiveness analysis, (3) quality of life assessment outcomes and (4) extensive safety data.

As this study is an open trial, in which the participants, clinicians and hospital staff will not be blinded to their treatment allocation, it is at risk of bias, particularly performance or observer bias. A control group undergoing a sham endoscopy would significantly reduce this bias but would expose a large number of patients to the risks of an unnecessary endoscopic procedure and general anaesthetic. Therefore, to reduce the effect of bias, (1) participants will undergo a concealed computer-generated randomisation process by an independent statistician, (2) multiple assessors across both study sites will follow structured assessment protocols and use validated measurement tools in order to minimise subjectivity from the data collection, (3) data collection will be monitored regularly to ensure adherence to the protocol and to perform source data verification, and (4) where possible, outcomes and results will be reported by an independent person who is unaware of the treatment allocation of the participant (eg, the primary outcome measure of HbA1c and all other haematological or biochemical samples will be measured and reported by an independent laboratory technician at each hospital). Attrition bias will be minimised by performing regularly scheduled follow-up visits across both treatment groups, and regular telephone follow-ups will be performed in order to assess the patient's well-being and motivation on the trial. Patients selected for this trial will be a very motivated subset of the population of interest. The effects of this sampling bias will be minimised through effective randomisation but will reduce the generalisability of any significant treatment effect identified.

To conclude, we hypothesise that exclusion of the foregut by means of an EndoBarrier device will improve glycaemic control, above that of conventional medical therapy, diet and exercise via (1) decreased hepatic insulin resistance and increased insulin production that occur independent of weight loss and caloric restriction, and (2) reduction in total body and tissue-specific insulin resistance as a result of consequent weight loss. We also hypothesise that this device will produce weight loss, above that of control patients, by reducing hunger, increasing satiety (therefore reducing food intake) and changing food preferences and hedonics away from high-energy sweet and fatty foods. If the EndoBarrier is effective at achieving long-lasting weight loss and glycaemic control, there is an obvious potential for health benefit and savings on future health and social care, through the avoidance of T2DM and related complications.

### Trial status
The trial opened for recruitment at Imperial College Healthcare NHS Trust in London on 18 November 2014 and then in University Hospital Southampton NHS Foundation Trust on 5 June 2015. Recruitment was completed across both sites on 18 October 2016 and all EndoBarrier devices were inserted by 23 January 2017. Participant follow-up continues across both sites with the anticipated trial completion date on 23 January 2019.

**Author affiliations**
[1]Southampton Biomedical Research Centre, University Hospital Southampton, Southampton, Hampshire, UK
[2]Imperial College Healthcare NHS Trust, St Mary's Hospital, London, UK
[3]Department of Public Health, Imperial College London, Imperial Clinical Trials Unit, London, UK
[4]PsychoNeuroEndocrinology Research Group, Neuropsychopharmacology Unit, Centre for Psychiatry and Computational, Cognitive and Clinical Neuroimaging Laboratory, Division of Brain Sciences, Imperial College London, London, UK
[5]Division of Diabetes, Endocrinology and Metabolic Medicine, Hammersmith Hospital, London, UK
[6]Southampton HTA Centre, University of Southampton, University of Southampton Science Park, Southampton, UK
[7]Computational Cognitive and Clinical Neuroimaging Group, Hammersmith Hospital, Imperial College London, London, UK
[8]Imperial Clinical Trials Unit, Imperial College London, London, UK
[9]School of Public Health, Imperial College London, London, UK
[10]Department of Investigative Medicine, Imperial College London, London, UK
[11]Diabetes Complications Research Centre, Conway Institute, Dublin, Ireland
[12]School of Medicine and Medical Sciences, University College Dublin, Dublin, Ireland
[13]Division of Computational and Systems Medicine, Department of Surgery and Cancer, Faculty of Medicine, Imperial College London, London, UK
[14]Department of Diabetes and Endocrinology, Southampton General Hospital, Southampton, UK
[15]Primary Care Medical Group, University of Southampton Medical School, Southampton, UK
[16]North West London Pathology, Head of Division of Diabetes, Endocrinology and Metabolism, Imperial College London, Hammersmith Hospital, London, UK
[17]Division of Surgery, Imperial College London, London, UK
[18]Head of Pathology, University College Dublin, Dublin, Ireland
[19]Division of Surgery, University Hospital Southampton NHS Foundation Trust, Southampton, UK
[20]Department of Gastroenterology, Imperial College Healthcare NHS Trust, St Mary's Hospital, London, UK

**Acknowledgements** Special thanks are given to Brunel University London and the Wolfson Centre for Translational Research, University of Surrey, for their support in sample analysis, and to Professors Desmond Johnston, Jeremy Nicholson and Elaine Holmes as coapplicants for the trial and for their valuable support in the set-up and design of this trial.

**Contributors** MAG: coinvestigator at University Hospital Southampton, corresponding and primary author of manuscript. AM: coinvestigator at Imperial College London, contributed to writing the manuscript and approved the final version. CGP: trial manager at Imperial College London and Imperial Clinical Trials Unit, coauthor of approved study protocol, contributed to writing the manuscript and approved the final version. APG: coinvestigator at Imperial College London and trial coapplicant, coauthor of approved study protocol, contributed to writing the manuscript and approved the final version. ADM: coinvestigator at Imperial College London, coauthor of approved study protocol, contributed to writing the manuscript and approved the final version. JL: coauthor of approved study protocol, contributed to writing the manuscript and approved the final version. NC: coinvestigator at Imperial College London, coauthor of approved study protocol, provided critical appraisal of current manuscript and approved the final version. EF: trial coapplicant, trial statistician, contributed to writing the manuscript and approved the final version. NAJ: trial statistician, contributed to writing the manuscript and approved the final version. WA-N: coinvestigator at Imperial College London, coauthor of approved study protocol, contributed to writing the manuscript and approved the final version. CS: provided critical appraisal of current manuscript and approved final version. JVL: contributed to approved study protocol, provided critical appraisal of the manuscript and approved the final version. MP: coinvestigator at University Hospital Southampton, contributed to approved study protocol, provided critical appraisal of the manuscript and approved the final version. ARA: trial coapplicant, contributed to trial set-up and design, provided critical appraisal of the manuscript and approved the final version. MM: trial coapplicant, contributed to trial set-up and design and was involved in the design of approved study protocol, provided critical appraisal of the manuscript and approved the final version. NP, SB, AD and CLR: trial coapplicants, contributed to trial set-up and design, provided critical appraisal of the manuscript and approved the final version. JPB: principal investigator at University Hospital Southampton, trial coapplicant, coauthor for approved study protocol, provided critical appraisal of the manuscript and approved the final version. JPT: chief investigator, trial coapplicant, coauthor for approved study protocol, provided critical appraisal of the manuscript and approved the final version.

**Funding** This project is funded by the Efficacy and Mechanism Evaluation (EME) Programme, an MRC and National Institute for Health Research (NIHR) partnership. EME reference 12/10/04. This paper presents independent research funded by the EME Programme and supported by the NIHR CRF and BRC at Imperial College Healthcare NHS Trust and University Hospital Southampton NHS Foundation Trust. The views expressed are those of the author(s) and not necessarily those of the EME Programme, the NHS, the NIHR or the Department of Health. This study is being executed with the support of GI Dynamics and with kind support of Nutricia Advanced Medical Nutrition for providing oral nutritional supplements.

**Competing interests** Dr Miras reports grants from Fractyl, personal fees from Novo Nordisk, and personal fees from Astra Zeneka outside the submitted work. Dr Goldstone reports funding supported by UK Medical Research Council and Wellcome Trust, outside of the submitted work. Professor Poulter's institution reports grants from HTA, during the conduct of the study. Dr le Roux reports grants from Science Foundation Ireland, grants from Health Research Board, during the conduct of the study; other from NovoNordisk, other from GI Dynamics, personal fees from Eli Lilly, grants and personal fees from Johnson and Johnson, personal fees from Sanofi Aventis, personal fees from Astra Zeneca, personal fees from Janssen, personal fees from Bristol-Myers Squibb, personal fees from Boehringer-Ingelheim, outside the submitted work.

**Ethics approval** Fulham Research Ethics Committee, London, (reference 14/LO/0871) on 10 July 2014.

**Provenance and peer review** Not commissioned; externally peer reviewed.

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
