## [Reviewer comments · BMJ Open]

ARTICLE DETAILS

TITLE (PROVISIONAL)	Study Protocol: A randomized controlled trial of a duodenal-jejunal bypass sleeve device (EndoBarrier®) compared with standard medical therapy for the management of obese subjects with type 2 diabetes mellitus.
AUTHORS	Glaysher, Michael; Mohanaruban, Aruchuna; Prechtel, Christina; Goldstone, Anthony; Miras, Alexander; Lord, Joanne; Chhina, Navpreet; Falaschetti, Emanuela; Johnson, Nicholas; Al-Najim, Werd; Smith, Claire; Li, Jia; Patel, Mayank; Ahmed, Ahmed; Moore, Michael; Poulter, Neil; Bloom, Stephen; Darzi, Ara; Le Roux, Carel; Byrne, James; Teare, Julian

VERSION 1 – REVIEW

REVIEWER	Carolina Casellini Eastern Virginia Medical School, USA
REVIEW RETURNED	28-Aug-2017

GENERAL COMMENTS	This is a well designed trial that will add important information on the effectiveness of liners in the treatment of diabetes, and will help to further understand the mechanisms by which bariatric surgery and liners improve the metabolic abnormalities seen in type 2 diabetes
---

REVIEWER	Gian Franco Adami Department of internal medicine Genova, Italy none
REVIEW RETURNED	12-Sep-2017

GENERAL COMMENTS	I believe that the primary end-point is not well defined. In fact the outcome was defined as percent reduction of glycated haemoglobin, this making a great difference between patient with a high basal Hba1c and their counterpart with Hba1c values only slightly above the normal range. I believe that in a study so interesting and well documented setting an absolute Hba1c value as target is mandatory.
---

VERSION 1 – AUTHOR RESPONSE

Response to Reviewer 2:

The primary end-point of a 20% reduction in HbA1c was chosen as the International Diabetes Federation (IDF) produced in June 2011 new guidelines for the conduct of studies in diabetes using bariatric surgery or devices with the intention of standardising outcomes and allowing comparison between studies.

dixon JB, Zimmet P,Alberti KG, Rubino F. Bariatric surgery: an IDF statement for obese Type 2 diabetes. Diabet Med. 2011 Jun; 28(6): 628–642

This working group reviewed all literature, national and international guidelines, systematic reviews and high-quality clinical trials published between 1991 and 2010 focusing on bariatric surgery for the treatment of obesity and diabetes in adults and adolescents. The group synthesized the available evidence for efficacy, safety and cost effectiveness of established bariatric procedures in relation to current standard therapy for people with obesity and Type 2 diabetes.

This international consensus concluded that an achievable goal of metabolic procedures (i.e. bariatric surgery or devices) is not cure, but remission, of the diabetic state. Along with the other parameters that form our secondary outcome measures (i.e. Blood pressure below 135/85 mmHg, Absolute weight loss greater than 15% etc), a reduction in HbA1C by 20% was deemed a clinical indicator of substantial improvement in a patients metabolic state. Optimization of the metabolic state was then defined by achieving a HbA1c < 42 mmol/mol or 6% (secondary outcome measure).

This primary end-point is therefore in line with national and international expert consensus. To date there are thus no published large patient group studies using this end-point, so using this new endpoint in a well-designed and conducted study will be of scientific value in itself.